# Regional Difference in the Effect of Food Accessibility and Affordability on Vegetable and Fruit Acquisition and Healthy Eating Behaviors for Older Adults

**DOI:** 10.3390/ijerph192214973

**Published:** 2022-11-14

**Authors:** Dong Eun Lee, Kirang Kim

**Affiliations:** Department of Food Science and Nutrition, Dankook University, Cheonan 31116, Korea

**Keywords:** food environment, vegetable, fruit, eating behavior, older adults

## Abstract

The food environment has been determined to affect a range of healthy eating and health indicators, but the study on the regional difference of food environment effects on these outcomes is limited. This study aimed to examine whether food environment factors influence vegetable and fruit acquisition and healthy eating behaviors in urban and rural areas using a nationwide dataset. The study participants were community-dwelling older adults aged 65 years and older (*n* = 830) who participated in the 2019 Consumer Behavior Survey for Food provided by the Korea Rural Economic Institute. Food environment factors were assessed using questionnaires measuring perceived food accessibility and affordability. The negative perceptions of food environment were related to lower vegetable and fruit acquisitions and poor healthy eating behaviors. The higher risks of low vegetable and fruit acquisitions in older rural adults were related to a negative perception of food accessibility only (odds ratio [OR]: 2.34, 95% confidence interval [CI]: 1.27–4.32 for vegetable; OR: 1.96, 95% CI: 1.02–3.75 for fruit). For older urban adults, negative perceptions of both food accessibility and food affordability were related to the increased risk of low vegetable acquisition (OR: 2.03, 95% CI: 1.07–3.83 for food accessibility; OR: 2.52, 95% CI: 1.26–5.04 for food affordability). In terms of healthy eating behaviors, for those who perceived that either food accessibility or affordability was poor, older urban adults were less likely to have various and healthy food eating behaviors when they had a negative perception of affordability (OR: 0.47, 95% CI: 0.25–0.90 for variety; OR: 0.23, 95% CI: 0.11–0.46 for eating healthy foods); however, older rural adults were less likely to have the behaviors when they had a negative perception of accessibility (OR: 0.49, 95% CI: 0.21–0.97 for variety; OR: 0.28, 95% CI: 0.13–0.63 for eating healthy foods). In conclusion, the negative perceptions of food accessibility and affordability were related to low vegetable acquisition and poor healthy eating behaviors. The effects of food accessibility and affordability on vegetable and fruit acquisitions and healthy eating behaviors were different between urban and rural areas.

## 1. Introduction

In 2021, the Korean older adult population was 16.5% of the total population, and it continues to increase, predicted to reach 20.3% in 2025 [1]. According to the results of the 2020 Population and Housing Census, the proportion of the adult population aged 65 years and older in urban and rural areas was 14.6% and 31.5%, respectively, indicating that the proportion of rural areas was twice that of urban areas [2]. A recent study reported that there was a critical inequality in the medical service quality and health status between rural and urban areas, showing poorer outcomes in rural areas [3].

Gaps between urban and rural areas for dietary quality were also noted. As a result of comparing the Korean healthy eating index by region and age in KNAHES data, the difference in the index score between areas was dependent on the age group. The difference was only significant for older adults aged 60 years or older [4]. The effect of region on dietary quality in older adults remained significant even after controlling for all individual component factors. The results of this study showed the possibility of the existence of structural environment factors that make a difference in the dietary quality in rural and urban areas. The area effect on dietary quality, as a contextual effect, could include the community food environment effect.

The food environment is a concept that includes physical, social, economic, cultural, and political factors that can affect food availability, accessibility, affordability, and adequacy in food retail and food service settings [5,6]. A healthy food environment can help support people in making healthier food choices and eating behaviors [7]. Several systemic review studies have reported that the availability and accessibility of grocery stores and food affordability were key factors of food environment and were associated to healthy food intakes [8,9,10]. Therefore, creating a healthy food environment is an important part of public nutritional policy. Certain strategies to improve the food environment have been suggested, for instance changing multiple settings such as home, work sites, school, restaurants, and supermarkets, social climate, information availability, and organizational systems to promote behavior change [7,11,12].

A recent study found different aspects of the food environment in urban and rural areas of three countries having different income levels and food systems and pointed out integrating strategies addressing the food environment in non-communicable disease-related health interventions [13]. Several studies have reported the different factors in explaining food choice behaviors in urban and rural areas. Personal economic resources, including home ownership and household income, were generally significant factors for a healthy diet in urban areas [14,15,16,17], and physical environment factors, including poor spatial accessibility to food stores, limited transportation system, and a lack of farming or gardening, were related to food intakes in rural areas [18,19,20,21]. A study found that the effect of the community food environment was stronger in rural areas than that in urban areas [22]. Therefore, it would be important to study the effect of food environments on healthy eating in different regional contexts because the food environment influences the food choice and intake of residents. Most studies have examined the findings in a single regional setting, and there are limited studies comparing the effect of food environment between urban and rural areas.

The intake of vegetable and fruit–which are known as key factors associated with positive health outcomes–is one of the target indicators of the National Health Plan 2030 in Korea [23]. According to the results of the 2020 National Health and Nutritional Examination Survey, the proportion of older adults who met the intake criteria–which is more than 400 g per day for vegetable and 100 g per day for fruit–was 43.8% for older men and 32.9% for older women [24]. As compared between areas, older rural adults had a higher risk of insufficient intake of vegetable and fruit than older urban adults [25]. As vegetable and fruit intake is affected by the ecological context of food choice, understanding the effect of the food environment on their consumption could help develop more fundamental strategies to increase vegetable and fruit intake among older adults. Therefore, using a nationwide dataset, this study aimed to examine whether food accessibility and affordability influence the vegetable and fruit acquisition and healthy eating behaviors of older adults living in urban and rural areas, and whether the effect is different between two areas.

## 2. Material and Methods

### 2.1. Data Source and Study Population

This study was based on the 2019 Consumer Behavior Survey for Food (CBSF) provided by the Korea Rural Economic Institute (KREI). These data were obtained from the CBSF website (https://www.krei.re.kr/foodSurvey/selectBbsNttView.do?key=1774&bbsNo=451&nttNo=132650, accessed on 7 November 2022). The CBSF is a nationwide cross-sectional survey conducted in 2013 to examine consumers’ perceptions and behaviors on their food consumption patterns and food purchase. The representative Korean adult household and household members aged 19–74 years old were collected using the stratified and multistage clustered probability sampling method. The study had a two-step survey. First, the household-level survey was designed to ask the primary food purchaser of a household in terms of the food consumption status and characteristics of the family. Second, the household member-level survey was designed to ask any household members (aged 19–74 years) about their food consumption status outside of home and their individual perceptions on food consumption. In this study, we used datasets for both household-level and household member-level surveys. Of the 6,176 individuals who participated in the survey, data were analyzed for 830, excluding participants under the age of 65 (*n* = 5319) and those with stroke, heart disease, or cancer (*n* = 27). All procedures and protocols used in the study were approved by the Institutional Review Board of Dankook University (DKU 2021-03-051). Written informed consent regarding the survey was obtained from all participants.

### 2.2. General Characteristics

The general characteristics of the survey participants were evaluated according to sex, age, educational levels, occupation, household type, average monthly income, government support program, owning a car, disease status, alcohol intake, regular exercise, and frequency of family eating out. Age was classified based on the median age of 70 among the older adults aged 65–75 years. The educational levels were classified into the following three groups: no education, middle school graduate or less, and high school graduate or higher. The household type was classified into living alone and living together. As the result of a survey by the National Statistical Office showed that the average income of the older adult household was 2.33 million Korean won (approximately 2000 US dollar), the average monthly income was classified into less than 2 million Korean won and more than 2 million Korean won [26].

Moreover, the participants were classified by whether or not there has been any experience in receiving benefits from the government support programs, such as basic livelihood security benefits, free meals and packed lunch delivery, and supplemental nutritional support program. Disease status was assessed by whether or not they have the following diseases: hypertension, hyperlipidemia, diabetes, thyroid disease, hepatitis, stomach and duodenal ulcer, and liver cirrhosis. Owning a car, drinking alcohol, and exercising regularly were classified into two groups according to whether or not they had owned a car, drunk alcohol, and exercised regularly. The frequency of family eating out was classified into three groups of less than one time per month, one to two times per month, and more than two times per month.

### 2.3. Food Environment Factors

We used two questions from the CBSF to measure food environment. These two questions were measured using a five-point Likert scale. The first question was “There are plenty of grocery stores close to home and is no physical difficulty in purchasing and preparing food.” The other question was “Our family can afford to purchase a sufficient amount and variety of food.” Each question was assessed as food accessibility and affordability, respectively, based on the components of food environment defined by Caspi et al. [8].

The responses to questions for food environment were scored from 1 (strongly disagree) to 5 (strongly agree) and were classified into two groups, defining them as negatively perceived food accessibility and affordability from 1 to 3 and positively perceived food accessibility and affordability from 4 to 5. Additionally, the combined groups for food accessibility and affordability were used to assess their relationship with healthy food acquisition and eating behaviors.

### 2.4. Vegetable and Fruit Acquisitions

The vegetable and fruit acquisition included both daily purchases and non-purchased sources such as own production or received foods for free or as aid. The frequency of vegetable and fruit acquisition included in the CBSF questionnaire was divided into seven categories, such as every day, 2–3 times a week, once a week, once every 2 weeks, once a month, rarely, and no intake of vegetables/fruits. This was subsequently categorized into less than once a week and once a week or more according to distribution of response. Additionally, the frequency of grocery purchases was measured as less than once a week and once a week or more. The place to purchase food and purchase foods online were also asked. For place to purchase food, local supermarkets operated by an individual were classified into small-size markets, local supermarkets operated by large corporations were medium-size markets, large discount stores (super supermarket) operated by large corporations were large-size markets, and local markets run by individual farmers were traditional markets.

### 2.5. Healthy Eating Behaviors

In terms of healthy eating behaviors, the CBSF assessed three components, including moderation, variety, and eating healthy foods. The components were measured using a five-point Likert scale for one question of each component. The moderation component was evaluated by asking “I do not overeat and eat as much as necessary.” The variety of food intake component was assessed by a question of “I eat a variety of foods for proper nutrition.” The healthy food intake component was measured by a question of “I usually eat a lot of vegetables, fruits, and whole grains.” The responses were scored from 1 (strongly disagree) to 5 (strongly agree) and classified into two groups, defining them as non-healthy eating behavior from 1 to 3 and healthy eating behavior from 4 to 5.

### 2.6. Statistical Analyses

The 2019 CBSF was analyzed using a complex sampling analysis that reflected the layer, cluster, and sampling weight with a complex sampling design. For categorical variables, frequencies and percentages were presented, and the statistical significance of differences between groups was tested using the chi-square test. To evaluate the relationship of food acquisition and healthy eating behaviors with food environment factors, the multiple logistic regression analysis was conducted to determine odds ratios (ORs) and 95% confidence intervals (95% CIs). All analyses were performed using SPSS Statistics 26 (IBM Company, Armonk, NY, USA). Statistical significance was defined at *p* < 0.05.

## 3. Results

### 3.1. General Characteristics of Study Subjects

The demographic and socioeconomic characteristics of study subjects are presented in Table 1. The proportion of older urban and rural adults was 57.7% and 42.3%, respectively; overall, 45.2% were males and 54.8% were females. The percentage of urban and rural areas aged 70 or older was 40.4% and 58.3%, respectively (*p* < 0.001). The urban areas were found to have higher educational levels than the rural areas, showing 48.1% in urban areas and 28.7% in rural areas for high school graduation or higher (*p* = 0.003). Urban areas had the highest proportion of unemployed or housewives, whereas rural areas had the highest proportion of agriculture, forestry, and fisheries (*p* < 0.001). The proportion of single household was 33.1% in urban order adults and 36.7% in rural adults, which was not significantly different. Most of older adults (92.8%) did not participate in the government support program. In terms of owning a car, older rural adults (55.9%) had a higher proportion than older urban adults (41.6%) (*p* = 0.03). About one-third of the subjects had a disease, 57.5% were drinkers, and 24.5% exercised regularly, which were not significantly different between urban and rural areas.

### 3.2. Food Acquisition, Healthy Eating Behavior, and Perceived Food Store Accessibility by Region

The comparison of food acquisition, healthy eating behaviors, and food environment by region is shown in Table 2. Differences were noted in the frequency of food purchases depending on the region, showing that older urban adults purchased foods more frequently (*p* = 0.021). Approximately two-thirds of older adults in urban and rural areas purchased foods from small-size markets or traditional markets. Overall, 5.8% of older adults purchased foods online, indicating that most of them did not use online when purchasing food, and when compared by region, the proportion of purchases through online was higher in urban areas (*p* = 0.005).

Regarding food acquisition, the rural areas had a lower frequency of vegetable acquisition than the urban areas (*p* = 0.046), which could be partly explained by higher direct growing in rural areas (*p* < 0.001). However, the frequency of fruit acquisition was not different between urban and rural areas (*p* = 0.26). Regarding healthy eating behaviors, the proportions of behaviors of various food intakes and eating healthy foods were higher in urban areas than those in rural areas (62.7% vs. 43.4% for various food intakes and 59.7% vs. 46.8% for eating healthy foods). Regarding food environment, the proportion of older adults with a positive perception of food accessibility was higher in urban areas than that in rural areas (59.5% vs. 47.2%, *p* = 0.03); however, the proportion of older adults with a negative perception of food affordability was not different between the two areas.

### 3.3. Perceived Food Environment and Acquisition of Vegetables and Fruits by Region

The relationship of food environment with vegetable and fruit acquisition by region is shown in Table 3. A difference in the frequency of vegetable acquisition according to the food environment but not in fruit acquisition was noted. Older adults with negative perceptions of the food environment had a relatively higher proportion of vegetable acquisition less than once a week than those with positive perceptions (27.3% vs. 15.1% for food accessibility [*p* = 0.001] and 25.1% vs. 15% for food affordability [*p* = 0.011]). Older adults with positive perceptions of both accessibility and affordability had a higher proportion of acquiring vegetables more frequently than those with negative perceptions of both of them (86.3% vs. 68% for once a week or more times [*p* = 0.001]).

The difference in food acquisition frequency by food environment was affected by region. The frequency of vegetable acquisition was more influenced by food affordability in urban areas and influenced by food accessibility in rural areas. The proportion of vegetable acquisition less than once a week was 21.3% in older urban adults with a negative perception of food affordability, whereas it was 10.8% in those with a positive perception of food affordability (*p* = 0.014). For older rural adults with a negative perception of food accessibility, the proportion of vegetable acquisition less than once a week was 34.9%, whereas it was 18% in those with a positive perception of accessibility (*p* = 0.014). Regarding fruit acquisition, older rural adults with a negative perception of food accessibility had a marginally significantly higher proportion of acquiring fruits less frequently than those with a positive perception of accessibility (45.3% vs. 29.9%, *p* = 0.052).

### 3.4. Perceived Food Environment and Healthy Eating Behaviors by Region

The relationship between food store accessibility and healthy eating behaviors by region is presented in Table 4. Generally, the proportion of healthy eating behavior was different by perceived food environment. The proportion of older adults with healthy eating behaviors was higher in those with positive perceptions of food environment than in those with negative perceptions (*p* < 0.001). In older urban adults, all proportions of healthy eating behaviors were higher in those with positive perceptions of both food accessibility and affordability than in those with negative perceptions. However, in older rural adults, the proportion of variety of food intakes and eating healthy foods was not different by perceived food affordability. 

### 3.5. The Effect of Perceived Food Environment on Low Vegetable and Fruit Acquisition by Region

The effect of the perceived food environment on food acquisition by region is presented in Table 5. When older adults perceived that food accessibility or affordability is poor, they had a higher risk of low vegetable acquisition after adjusting for confounding variables (OR: 2.15, 95% CI: 1.40–3.30 for food accessibility; OR: 1.86, 95% CI: 1.16–2.97 for food affordability). The effect of both food accessibility and affordability on vegetable acquisition was significant in older urban adults (OR: 2.03, 95% CI: 1.07–3.83 for food accessibility; OR: 2.52, 95% CI: 1.26–5.04 for food affordability). However, for older rural adults, only food accessibility was associated with a higher risk of low vegetable acquisition (OR: 2.34, 95% CI: 1.27–4.32). Regarding low fruit acquisition, food accessibility was associated with a higher risk in only older rural adults (OR: 1.96, 95% CI: 1.02–3.75). Older adults with negative perceptions of both food accessibility and affordability had a higher risk of low vegetable acquisition than those who had positive perceptions of both food accessibility and affordability. However, the higher risk was not significant for those with a negative perception of either of them.

### 3.6. Effect of Perceived Food Store Accessibility on Healthy Eating Behaviors by Region

The effect of the perceived food environment on healthy eating behaviors by region is shown in Table 6. When older adults perceived that food accessibility or affordability is poor, they were less likely to have all healthy eating behaviors after adjusting for confounding variables (OR: 0.33, 95% CI: 0.21–0.52 for moderation; OR: 0.42, 95% CI: 0.27–0.63 for variety; and OR: 0.28, 95% CI: 0.19–0.41 for eating healthy foods). The positive effect of food accessibility or affordability on healthy eating behaviors was significant in both urban and rural older adults except for the finding of variety in older rural adults. 

Older adults with negative perceptions of both food accessibility and affordability were less likely to have healthy eating behaviors than those with positive perceptions of both food accessibility and affordability. For those with a negative perception of either of them, the risk of each healthy eating behavior was different by region. Older uban adults were less likely to have various and healthy food eating when they had a negative perception of food affordability and a positive perception of food accessibility than vice versa (OR: 0.47, 95% CI: 0.25–0.90 for variety; OR: 0.23, 95% CI: 0.11–0.46 for eating healthy foods); however, older rural adults were less likely to have the behaviors when they had a negative perception of food accessibility and a positive perception of food affordability than vice versa (OR: 0.49, 95% CI: 0.21–0.97 for variety; OR: 0.28, 95% CI: 0.13–0.63 for eating healthy foods).

## 4. Discussion

The nutritional disparity between urban and rural older adult populations has been underscored in Korea [4,27]. The regional dietary disparity could be explained by personal factors living in the areas and community food environment affecting accessibility to healthy foods [8,18,28,29]. As the effect of the food environment on healthy food intakes has been known in previous studies [8], this study aimed to examine how the food environment affects healthy food acquisition in different contexts of urban and rural areas of Korea, and whether the effect differs in these areas. This study found that negative perceptions of the food environment were related to low vegetable and fruit acquisition and poor healthy eating behaviors. The dimension of the food environment influencing vegetable and fruit acquisition and healthy eating behaviors differed according to region. For the older rural adults, a negative perception of food accessibility was inversely related to frequent vegetable and fruit acquisitions and healthy eating behaviors, whereas a negative perception of food affordability was not. For the older urban adults, those with a negative perception of food affordability were at higher risk of low vegetable acquisition than those with a negative perception of food accessibility. In addition, older urban adults who perceived food affordability negatively and food accessibility positively were less likely to have various and healthy food eating than those who perceived it to be the opposite.

Understanding context-specific factors to enable the older adult population to acquire their food is crucial to develop and implement effective interventions. The effect of the local food environments on food intakes could depend on distinct characters in urban and rural ecological contexts. Several previous studies on the association of the food environment with healthy diets found that the effect of food environment on healthy diets was stronger in rural areas than that in urban areas owing to poorer spatial accessibility of food stores and inadequate household food resources in rural areas [22,30]. Similar results were also shown in previous Korean studies [31,32]. The nationwide study on the food purchase and dietary habits of households across the country showed that the rural population had difficulty in accessing food or that there were not enough grocery stores to purchase food than the urban population [31]. This study showed that a negative perception of food accessibility in older rural adults was inversely related to frequent vegetable and fruit acquisitions and healthy eating behaviors, but a negative perception of food affordability was not. As public transportation is particularly lacking in rural areas, transportation may be problematic among older rural adults, forcing those who do not have their own vehicles or cannot drive to rely on family members, friends, and others for their transportation or shopping [9,13]. Therefore, delivery services or mobile markets would be tailored to overcome rural older adults’ specific food accessibility needs.

Conversely, some studies have reported that spatial accessibility of grocery stores was not related to healthy diets [14,17,33,34] These results were more significant in urban areas than in rural areas. In particular, the density of grocery stores is very high in urban areas of Korea, expecting easy physical access to local grocery stores. Thus, the environmental effect of physical distance would have a low effect on food purchases in urban older adults in Korea [17]. On the contrary, among the older adults in the urban area, food affordability would be the most significant factor in the food environment that can affect the food choices of the economically vulnerable older adults The price of food in the community is well known to influence healthy food intake in both urban and rural food environments [35]. In this study, a negative perception of food affordability was related to a low vegetable acquisition in urban older adults but not in rural older adults. Compared with older urban adults, the reason that food affordability did not affect vegetable acquisition in older rural adults could be explained by the higher percentage of vegetable self-sufficiency by farming in rural areas. Therefore, providing food or cash assistance services to address low food affordability needs in urban older adults would be useful.

This study found that the food environment affected healthy eating behaviors focusing on moderation, variety, and eating healthy foods, showing that a supportive food environment with easily accessible healthy foods may provide an opportunity to change to healthy dietary behaviors. A different effect of each food accessibility and affordability on the behaviors by regions was noted, which was consistent with their effect on healthy food acquisition. Although the effect of the food environment on eating behaviors could be mediated by that of healthy food acquisition, further study of mediation analysis focusing on identifying the mechanisms through which interventions have an effect may help design more efficient and effective interventions in various regional contexts.

This study had some limitations. First, the causal direction of the relationships of food environment with healthy food acquisition and healthy eating behaviors were unknown owing to the cross-sectional design. Second, the methodological weaknesses of this study stemmed from the subjective two-item measures of the food environment, which was not validated. In the future, the effect of the regional food environment on dietary quality with extra dimensions of food environmental variables related to the characteristics of the community should be studied. Finally, other plausible factors related to food acquisition or eating behaviors, including social network or psychological factors, could not be considered for the analysis due to unavailable information. Despite several limitations, this study emphasized the significance of the food environments in the design of interventions across urban and rural food environments using nationwide representative data. Food choice or eating behavior is a behavior that occurs within an ecological context consisting of distinct characters in urban and rural food environments. Therefore, the intervention should account for the difference in the context of the regional food environment.

## 5. Conclusions

Our findings showed that negative perceptions of food accessibility and affordability were related to lower vegetable acquisition and poor healthy eating behaviors. The effects of two food environment dimensions on vegetable and fruit acquisitions and healthy eating behaviors were different between urban and rural areas. For older rural adults, low vegetable and fruit acquisitions and poor healthy eating behaviors were significantly related to a negative perception of food accessibility, whereas for older urban adults, the low vegetable acquisition and poor healthy eating behaviors were significantly related to a negative perception of food affordability. These findings would be meaningful in developing a policy intervention strategy from a macro perspective, making food environments more conducive to healthy choices.

## Figures and Tables

**Table 1 ijerph-19-14973-t001:** General characteristics of study subjects.

Variable	Total (*n =* 830)		Region				
			Urban (*n =* 479)		Rural (*n =* 351)		*p* Value *
	*n*	%	*n*	%	*n*	%	
Sex							
Male	348	45.2	208	45.6	140	44.6	0.810
Female	482	54.8	271	54.4	211	55.4	
Age, years							
<70	491	52.4	308	59.6	183	41.7	<0.001
≥70	339	47.6	171	40.4	168	58.3	
Educational level							
Not attending school	56	6.2	30	5.7	26	7.0	0.003
Middle school	479	53.5	239	46.3	240	64.3	
≥High school	295	40.3	210	48.1	85	28.7	
Occupation							
Administrator/Professional/Sales/Service	131	13.9	100	18.9	31	6.4	<0.001
Agriculture/Forestry/Fisheries	228	24.7	23	6.0	205	52.6	
Technician	163	23.4	126	29.2	37	14.9	
Housewife, Unemployed	308	38.0	230	45.9	78	26.1	
Household types							
Single	231	34.5	129	33.1	102	36.7	0.515
Non-single	599	65.5	350	66.9	249	63.3	
Household income (thousand won)							
<2000	469	58.0	262	55.1	207	62.3	0.205
≥2000	361	42.0	217	44.9	144	37.7	
Government support program							
Yes	63	7.2	23	7.0	40	7.5	0.854
No	767	92.8	456	93.0	311	92.5	
Owning a car							
Yes	421	47.3	219	41.6	202	55.9	0.030
No	409	52.7	260	58.4	149	44.1	
Disease							
Yes	265	31.9	158	34.0	107	28.8	0.240
No	565	68.1	321	66.0	244	71.2	
Alcohol consumption							
Yes	455	57.5	263	56.3	192	59.2	0.584
No	375	42.5	216	43.7	159	40.8	
Regular exercise							
Yes	198	24.5	126	26.0	72	22.1	0.390
No	632	75.5	353	74.0	279	77.9	
Frequency of family eating out							
<1 time/month	339	36.8	187	36.6	152	37.1	0.967
1–2 times/month	377	47.2	227	47.0	150	47.5	
>2 times/month	114	16.0	65	16.4	49	15.3	

All percentages are calculated by applying sampling weights. * *p* values for percentage differences between the two groups are calculated using the χ2 test.

**Table 2 ijerph-19-14973-t002:** Characteristics of food acquisition, healthy eating behaviors, and perceived food environment by region.

Variable	Total		Region				*p* Value *
			Urban		Rural		
	*n*	%	*n*	%	*n*	%	
Food acquisition							
Total food purchase frequency							
<1 times/week	175	79.7	72	15.3	103	27.8	0.021
≥1 times/week	655	20.3	407	84.7	248	72.2	
Food purchase place							
Small-size market	300	37.9	167	36.5	133	39.8	0.408
Medium-size market	104	15.5	60	18.0	44	11.7	
Large-size market	145	15.3	84	13.3	61	18.4	
Traditional market	281	31.3	168	32.2	113	30.1	
Purchase food online							
Yes	53	5.8	41	8.1	12	2.4	0.005
No	777	94.2	438	91.9	339	97.6	
Vegetable acquisition							
Frequency							
<1 time/week	184	20.6	85	16.4	99	26.9	0.046
≥1 time/week	646	79.4	394	83.6	252	73.1	
Type							
Direct cultivation	226	22.7	56	11.2	170	40.0	<0.001
Purchase or acquisition from relatives	600	77.3	422	88.8	178	60.0	
Fruits acquisition							
Frequency							
<1 time/week	311	34.1	153	31.5	158	38.0	0.260
≥1 time/week	519	65.9	326	68.5	193	62.0	
Type							
Direct cultivation	11	0.7	0	0.0	11	1.6	<0.001
Purchase or acquisition from relatives	814	99.3	476	100.0	338	98.4	
Healthy eating behavior							
Moderation							
No	311	38.0	172	35.7	139	41.3	0.360
Yes	519	62.0	307	64.3	212	58.7	
Variety							
No	375	45.0	188	37.3	187	56.6	0.001
Yes	455	55.0	291	62.7	164	43.4	
Eating healthy foods							
No	357	45.5	182	40.3	175	53.2	0.018
Yes	473	54.5	297	59.7	176	46.8	
Food environment							
Food accessibility (AC)							0.030
Positive	466	54.6	285	59.5	181	47.2	
Negative	364	45.4	194	40.5	170	52.8	
Food affordability (AF)							0.354
Positive	389	44.8	232	46.9	157	41.6	
Negative	441	55.2	247	53.1	194	58.4	
AC and AF							0.084
Positive AC and AF	248	29.8	159	34.3	89	23.0	
Positive AC and negative AF	218	24.8	126	25.2	92	24.2	
Negative AC and positive AF	141	15.0	73	12.6	68	18.6	
Negative AC and AF	223	30.4	121	27.9	102	34.2	

All percentages are calculated by applying sampling weights. * *p* values for percentage differences between the two groups are calculated using the χ2 test.

**Table 3 ijerph-19-14973-t003:** The relationship of perceived food environment with vegetable and fruit acquisition.

	Food Accessibility				Food Affordability				Food Accessibility (AC) and Food Affordability (AF)							
	Positive		Negative		Positive		Negative		Positive AC and AF		Positive AC and Negative AF		Negative AC and Positive AF		Negative AC and AF	
	*n*	%	*n*	%	*n*	%	*n*	%	*n*	%	*n*	%	*n*	%	*n*	%
Total																
Vegetables																
<1 time/week	91	15.1	93	27.3	72	15.0	112	25.1	43	13.7	48	16.7	29	17.7	64	32.0
≥1 time/week	375	84.9	271	72.7	317	85.0	329	74.9	205	86.3	170	83.3	112	82.3	159	68.0
*p* value *	0.001				0.011				0.001							
Fruits																
<1 time/week	159	31.0	152	37.9	130	33.3	181	34.8	79	30.6	80	31.5	51	38.6	101	37.6
≥1 time/week	307	69.0	212	62.1	259	66.7	260	65.2	169	69.4	138	68.5	90	61.4	122	62.4
*p* value *	0.127				0.733				0.484							
Urban																
Vegetables																
<1 time/week	47	13.5	38	20.6	32	10.8	53	21.3	23	11.1	24	16.7	9	10.0	29	25.4
≥1 time/week	238	86.5	156	79.4	200	89.2	194	78.7	136	88.9	102	83.3	64	90.0	92	74.6
*p* value *	0.103				0.014				0.031							
Fruits																
<1 time/week	91	31.6	62	31.4	67	30.1	86	32.7	46	29.4	45	34.5	21	32.1	41	31.1
≥1 time/week	194	68.4	132	68.6	165	69.9	161	67.3	113	70.6	81	65.5	52	67.9	80	68.9
*p* value *	0.980				0.642				0.910							
Rural																
Vegetables																
<1 time/week	44	18.0	55	34.9	40	22.1	59	30.3	20	19.5	24	16.6	20	25.4	35	40.1
≥1 time/week	137	82.0	115	65.1	117	77.9	135	69.7	69	80.5	68	83.4	48	74.6	67	59.9
*p* value *	0.014				0.283				0.032							
Fruits																
<1 time/week	68	29.9	90	45.3	63	38.5	95	37.7	33	33.2	35	26.8	30	45.1	60	45.4
≥1 time/week	113	70.1	80	54.7	94	61.5	99	62.3	56	66.8	57	73.2	38	54.9	42	54.6
*p* value *	0.052				0.909				0.216							

All percentages are calculated by applying sampling weights. * *p* values for percentage differences between the two groups are calculated using the χ2 test.

**Table 4 ijerph-19-14973-t004:** The relationship of perceived food environment with healthy eating behaviors.

	Food Accessibility				Food Affordability				Food Accessibility (AC) and Food Affordability (AF)							
	Positive		Negative		Positive		Negative		Positive AC and AF		Positive AC and Negative AF		Negative AC and Positive AF		Negative AC and AF	
	*n*	%	*n*	%	*n*	%	*n*	%	*n*	%	*n*	%	*n*	%	*n*	%
Total																
Moderation																
No	140	26.2	171	52.1	104	23.8	207	49.5	60	21.4	80	32.0	44	28.5	127	63.8
Yes	326	73.8	193	47.9	285	76.2	234	50.5	188	78.6	138	68.0	97	71.5	96	36.2
*p* value *	<0.001				<0.001				<0.001							
Variety																
No	173	34.8	202	57.4	143	36.5	232	52.0	79	30.0	94	40.5	64	49.2	138	61.4
Yes	293	65.2	162	42.6	246	63.5	209	48.0	169	70.0	124	59.5	77	50.8	85	38.6
*p* value *	<0.001				0.001				<0.001							
Eating healthy foods																
No	155	31.2	202	62.6	104	29.4	253	58.5	51	20.9	104	43.6	53	46.3	149	70.6
Yes	311	68.8	162	37.4	285	70.6	188	41.5	197	79.1	114	56.4	88	53.7	74	29.4
*p* value *	<0.001				<0.001				<0.001							
Urban																
Moderation																
No	81	24.6	91	52.2	60	21.9	112	48.0	36	18.7	45	32.5	24	30.5	67	62.0
Yes	204	75.4	103	47.8	172	78.1	135	52.0	123	81.3	81	67.5	49	69.5	54	38.0
*p* value *	<0.001				<0.001				<0.001							
Variety																
No	94	29.9	94	48.2	64	26.0	124	47.4	39	22.9	55	39.4	25	34.2	69	54.6
Yes	191	70.1	100	51.8	168	74.0	123	52.6	120	77.1	71	60.6	48	65.8	52	45.4
*p* value *	0.003				<0.001				<0.001							
Eating healthy foods																
No	85	29.0	97	56.9	44	20.3	138	57.9	25	16.3	60	46.2	19	31.3	78	68.4
Yes	200	71.0	97	43.1	188	79.7	109	42.1	134	83.7	66	53.8	54	68.7	43	31.6
*p* value *	<0.001				<0.001				<0.001							
Rural																
Moderation																
No	59	29.3	80	52.1	44	27.0	95	51.6	24	27.2	35	31.2	20	26.6	60	66.0
Yes	122	70.7	90	47.9	113	73.0	99	48.4	65	72.8	57	68.8	48	73.4	42	34.0
*p* value *	0.009				<0.001				<0.001							
Variety																
No	79	44.0	108	67.8	79	54.1	108	58.3	40	45.8	39	42.2	39	64.4	69	69.7
Yes	102	56.0	62	32.2	78	45.9	86	41.7	49	54.2	53	57.8	29	35.6	33	30.3
*p* value *	0.006				0.610				0.023							
Eating healthy foods																
No	70	35.4	105	69.1	60	44.6	115	59.3	26	30.9	44	39.6	34	61.5	71	73.2
Yes	111	64.6	65	30.9	97	55.4	79	40.7	63	69.1	48	60.4	34	38.5	31	26.8
*p* value *	<0.001				0.079				<0.001							

All percentages are calculated by applying sampling weights. * *p* values for percentage differences between the two groups are calculated using the χ2 test.

**Table 5 ijerph-19-14973-t005:** The effect of perceived food environment on low vegetable and fruit acquisition by region.

	Vegetables						Fruits					
	Total		Urban		Rural		Total		Urban		Rural	
	OR *	95% CI	OR *	95% CI	OR *	95% CI	OR *	95% CI	OR *	95% CI	OR *	95% CI
Food accessibility												
Positive	1.00	Ref.	1.00	Ref.	1.00	Ref.	1.00	Ref.	1.00	Ref.	1.00	Ref.
Negative	2.15	1.40, 3.30	2.03	1.07, 3.83	2.34	1.27, 4.32	1.31	0.88, 1.97	1.00	0.61, 1.66	1.96	1.02, 3.75
Food affordability												
Positive	1.00	Ref.	1.00	Ref.	1.00	Ref.	1.00	Ref.	1.00	Ref.	1.00	Ref.
Negative	1.86	1.16, 2.97	2.52	1.26, 5.04	1.73	0.86, 3.47	1.04	0.69, 1.58	1.03	0.61, 1.74	1.20	0.65, 2.22
Food accessibility (AC) and Food affordability (AF)												
Positive AC and AF	1.00	Ref.	1.00	Ref.	1.00	Ref.	1.00	Ref.	1.00	Ref.	1.00	Ref.
Positive AC and negative AF	1.13	0.60, 2.14	1.43	0.63, 3.24	1.17	0.40, 3.42	0.93	0.53, 1.65	0.90	0.45, 1.82	1.08	0.41, 2.85
Negative AC and positive AF	1.23	0.62, 2.45	0.77	0.23, 2.66	1.72	0.75, 3.98	1.23	0.67, 2.24	0.81	0.38, 1.72	1.97	0.74, 5.23
Negative AC and AF	3.00	1.66, 5.42	3.57	1.50, 8.50	3.03	1.27, 7.23	1.29	0.75, 2.24	1.04	0.54, 2.01	2.05	0.88, 4.79

Ref., reference category. OR and 95% CI are calculated by applying sampling weights. * OR and 95% confidence interval are obtained using multiple logistic regression analysis after adjusting for sex, age, educational level, occupation, household types, household income, disease status, frequency of family eating out, alcohol consumption, regular exercise, and owning a car.

**Table 6 ijerph-19-14973-t006:** The effect of perceived food environment on healthy eating behaviors by region.

	Moderation						Variety						Eating Healthy Foods					
	Total		Urban		Rural		Total		Urban		Rural		Total		Urban		Rural	
	OR *	95% CI	OR *	95% CI	OR *	95% CI	OR *	95% CI	OR *	95% CI	OR *	95% CI	OR *	95% CI	OR *	95% CI	OR *	95% CI
Food accessibility																		
Positive	1.00	Ref.	1.00	Ref.	1.00	Ref.	1.00	Ref.	1.00	Ref.	1.00	Ref.	1.00	Ref.	1.00	Ref.	1.00	Ref.
Negative	0.33	0.21, 0.52	0.28	0.15, 0.49	0.41	0.20, 0.81	0.42	0.27, 0.63	0.45	0.26, 0.77	0.47	0.23, 0.94	0.28	0.19, 0.41	0.29	0.17, 0.49	0.28	0.15, 0.52
Food affordability																		
Positive	1.00	Ref.	1.00	Ref.	1.00	Ref.	1.00	Ref.	1.00	Ref.	1.00	Ref.	1.00	Ref.	1.00	Ref.	1.00	Ref.
Negative	0.29	0.20, 0.43	0.29	0.17, 0.49	0.25	0.14, 0.45	0.54	0.36, 0.79	0.39	0.23, 0.66	0.67	0.37, 1.19	0.28	0.19, 0.43	0.18	0.10, 0.32	0.44	0.23, 0.82
Food accessibility (AC) and Food affordability (AF)																		
Positive AC and AF	1.00	Ref.	1.00	Ref.	1.00	Ref.	1.00	Ref.	1.00	Ref.	1.00	Ref.	1.00	Ref.	1.00	Ref.	1.00	Ref.
Positive AC and negative AF	0.49	0.30, 0.80	0.44	0.22, 0.85	0.57	0.27, 1.20	0.65	0.40, 1.06	0.47	0.25, 0.90	0.76	0.35, 1.64	0.32	0.18, 0.57	0.23	0.11, 0.46	0.48	0.18, 1.31
Negative AC and positive AF	0.62	0.34, 1.13	0.45	0.21, 0.98	1.04	0.39, 2.81	0.48	0.26, 0.87	0.58	0.25, 1.36	0.49	0.21, 0.97	0.31	0.17, 0.56	0.40	0.16, 1.67	0.28	0.13, 0.63
Negative AC and AF	0.15	0.09, 0.26	0.13	0.06, 0.27	0.15	0.07, 0.36	0.29	0.17, 0.49	0.25	0.12, 0.50	0.37	0.16, 0.89	0.11	0.06, 0.19	0.08	0.04, 0.18	0.16	0.07, 0.37

Ref., reference category. OR and 95% CI are calculated by applying sampling weights. * OR and 95% confidence interval are obtained using multiple logistic regression analysis after adjusting for sex, age, educational level, occupation, household types, household income, disease status, frequency of family eating out, alcohol consumption, regular exercise, and owning a car.

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
