# Peer review of "Regional Difference in the Effect of Food Accessibility and Affordability on Vegetable and Fruit Acquisition and Healthy Eating Behaviors for Older Adults"

_ijerph, 2022, doi:10.3390/ijerph192214973_

Round 1

Reviewer 1 Report (Previous Reviewer 1)

The manuscript has been revised. Please make an editorial correction, in particular in Table 1 (percentage results). Please refer to Table 2 for an explanation of the abbreviations used.

Reviewer 2 Report (Previous Reviewer 2)

Dear Authors,

Comments were satisfactorily addressed. Congratulations on your contribution.

This manuscript is a resubmission of an earlier submission. The following is a list of the peer review reports and author responses from that submission.

Round 1

Reviewer 1 Report

Review of the article: Regional difference in the effect of food accessibility and affordability on vegetable and fruit acquisition and healthy eating behaviors for older adults

The idea of this study is good. The manuscript is attractive and well planned. However, this article needs some clarification and correction.

Major comments

·       Please explain why only fruit and vegetables were selected for analysis

·       In the abstract, please specify the meaning of the region - the division into rural and urban

·       References are very few

Chapter – Introduction

·       It would be worth supplementing the information about the habitual diet of Koreans, particularly regarding the amount of fruit and vegetables consumed, and describing how this diet is characterized.

·       Please explain why only fruit and vegetables were selected for analysis

Chapter – Material and Methods

·       Why did the exclusion criteria not consider other diseases affecting mobility and product selection?

Chapter – Results

·       Is it possible to present the average energy value and basic microelements for the rural and urban groups?

Chapter – Discussion

Discuss your results compared to other studies.

Chapter – Conclusions

Conclusions should be more clarificated.

Minor comments

·       Line 167: Make an editorial correction. In Table 1, improve the way of presenting the percentage results.

·        Please correct tables 2 and 4 to make them more readable.

Reviewer 2 Report

This paper investigated the regional difference in food accessibility and affordability on vegetable and fruit acquisition and healthy eating behaviors for older adults. The topic of study is interesting and has its own special scientific significance. However, some parts of the article need to be completely edited. I have the following comments.

 Abstract:

Line 8: Before stating the purpose of the study, in the background section, the importance of food environment and the factors affecting it should be mentioned.

Line 12: Please specify what food environmental factors have been evaluated?

Line 13: What do you mean by insufficient food environment? Maybe you mean insufficient food accessibility. If so, it should be corrected in the whole text, and result and discussion sections.

Line 23-24: In conclusion, please point to the important factors of the food environment that have an effect on the food acquisition and healthy eating behaviors in older adults.

 Keywords

- It is recommended that Keywords should be different from the title.

Introduction:

In general, in the introduction section, the authors should pay attention to the following questions and include them in the text of the article.

- What is the importance of food environment?

- What are food environmental factors? (Line 45-46)

- How can we improve the food environment?

- What is a healthy eating environment?

- What is the relationship between food and the environment?

- What are the strategies that can contribute to a healthy food environment?

- What are the important regional factors that affect the food acquisition by older adults in urban and rural community?

- Line 33: Please omit one of the "that"s.

-The introduction section should be stronger. More recent studies regarding the above questions should be reviewed by the authors.

- The objectives of the study should be explained more precisely.

 Materials and methods:

Section 2.1: Please indicate the method used to complete the questionnaires.

- Please provide the questionnaires prepared and used in the study as a supplementary file.

- How have you checked the validity and reliability of the questionnaires?

 Results and discussion

- The result and discussion section in this paper is very important and the authors should present and explain the results obtained in their study well and compare it with the results of others. Therefore, this part should be strengthened.

- In all the obtained results, the amount of ± SD should be determined and added to the results.

- In the discussion section, it should be explained about the regression analysis obtained in the consumption of fruits and vegetables in the studied urban and rural community.

- The conclusion must be clear, and concise, and based on the results obtained in the study.

Reviewer 3 Report

The work is focused on the study of the food environment effect (food accessibility and affordability) on healthy eating behaviours and vegetable and fruit acquisition of older adults. A deep study from the results of the survey evaluated has been carried out with a good statistical treatment. As a result, the study determined the main differences between rural and urban regions in food environment which conditionate the food decisions and actions. Taking all of these results into account it would be possible to make decisions and strategy plans to improve the social intervention depending on the region of people targeted.

Therefore, I recommend this article to be published in this journal after minor revision.

Abstract:

This section is concise and well structured; however, it is hard to understand from line 17 to 19, rephrase to make it more clear.

Introduction:

-        Line 32: Although it is mentioned in the text later, define here the age of older adults in order to know what population has been studied.

-        Line 33-34: “that” is mentioned twice, and “of” is not necessary.

-        Line 45: Explain more deeply the food environment concept.

-    Last paragraph: In the objective of the work, it should be mentioned the target population of study (older adult). Justify why has that population been chosen for the study.

Material and Methods:

-        Section 2.1.: In this section it is mentioned that household members aged 19-74 years old have been selected for the survey (line 77); however, later (line 82) it is said that in the second step of the survey, the household members asked were from 13 to 74 years. Please check.

-        Line 90: Add “Survey participants” general characteristics to make it more clear.

-        Line 109: Define “regularly” for exercise practice.

-    Section 2.4.: Explain why the frequency of vegetables and fruit acquisition was divided into seven categories if later it was just categorized in two sections. 

Results:

-        Section 3.1.: It would be interesting to highlight that there were not found significant differences for the rest of the variables (p ≤ 0.05).

-        Line 164: Correct “p<0.0.01”

-        Line 174: Percentage of small-size markets together with the traditional markets is almost 70 % (69.2 %), not one-third as it is said.

-        Line 178: p is not 0.006 but 0.005.

-        Table 2: It should be explained in the text (2.4. section) the difference between “small-”, “medium-”, “large-size markets” and “traditional markets”. 

-        Line 182-183: Underline that the result for fruit acquisition was not significant (p=0.260).

-        Line 186: It is not 56.6% but 43.4% neither 53.2% but 46.8%.

Conclusion:

This section is very brief. Point out the concrete results achieved in the study.